# Ecology and Infection Status of Sand Flies in Rural and Urban Cutaneous Leishmaniasis Endemic Areas in Northwest Ethiopia

**DOI:** 10.3390/tropicalmed9030052

**Published:** 2024-02-23

**Authors:** Wondmeneh Jemberie, Abebe Animut, Sisay Dugassa, Araya Gebresilassie, Roma Melkamu, Esayas Aklilu, Mulugeta Aemero, Johan van Griensven, Myrthe Pareyn

**Affiliations:** 1Vector Biology & Control Research Unit, Aklilu Lemma Institute of Pathobiology, Addis Ababa University, Addis Ababa P.O. Box 1176, Ethiopia; abebe.animut@aau.edu.et (A.A.); sisay.dugassa@aau.edu.et (S.D.); esayas.aklilu@aau.edu.et (E.A.); 2Department of Biology, College of Natural and Computational Sciences, University of Gondar, Gondar P.O. Box 196, Ethiopia; 3Department of Zoological Sciences, College of Natural and Computational Sciences, Addis Ababa University, Addis Ababa P.O. Box 1176, Ethiopia; araya.gebresilassie@aau.edu.et; 4Leishmania Research and Treatment Center, University of Gondar, Gondar P.O. Box 196, Ethiopia; roma.melkamu@uog.edu.et; 5Department of Medical Parasitology, School of Biomedical & Laboratory Sciences, College of Medicine & Health Sciences, University of Gondar, Gondar P.O. Box 196, Ethiopia; mulugeta.aemero@uog.edu.et; 6Clinical Sciences Department, Institute of Tropical Medicine Antwerp, 2000 Antwerp, Belgium; jvangriensven@itg.be (J.v.G.); myrthepareyn@itg.be (M.P.)

**Keywords:** cutaneous leishmaniasis, sand flies, *Phlebotomus longipes*, *Leishmania aethiopica*, Ethiopia

## Abstract

Cutaneous leishmaniasis (CL) caused by *Leishmania aethiopica* is transmitted by *Phlebotomus longipes* in northern Ethiopia. No studies have been conducted to investigate the transmission dynamics of CL, despite its high endemicity in both rural and urban settings. Evidence on the ecology and behavior of the vector from this area are required to develop integrated disease control strategies. Sand flies were collected in the dry and wet seasons in 2021 in CL-endemic rural Gindmeteaye and urban Addis-Alem in northwest Ethiopia. Trapping was performed with sticky and Centers for Disease Control and Prevention (CDC) light traps in three habitats, including inside patients’ houses, peridomestic areasand in caves/rocky areas. Sand flies were morphologically identified to species level. Female *Phlebotomus* species were categorized according to blood feeding status and tested by spliced-leader (SL-) ribonucleic acid (RNA) polymerase chain reaction (PCR) to screen for *Leishmania* infection. Of 1161 sand flies, the majority (77%) were *P. longipes*, six (0.5%) were *P. orientalis* and the remaining were *Sergentomyia*. The abundance of the 430 female *P. longipes* was significantly linked to seasonality (*p* < 0.001), with the majority in the dry season occurring in the outdoor rocky (37%) and peridomestic (34%) sites, while, in the wet season, most (62%) were captured indoors. This seasonality was more pronounced in rural Gindmeteaye, where housing construction is poor. The number of blood-fed and gravid *P. longipes* was significantly higher in the wet (31%; 22%), compared to the dry season (13%; 8%), and their proportion was highest indoors. Eighteen (4%) female *P. longipes* were *Leishmania* positive, with highest infection prevalence in caves (7% compared to 3% indoors, *p* = 0.022), and in the dry season (6%, *p* < 0.001). *Phlebotomus orientalis* specimens were all captured in May in rural Gindmeteaye, five indoors and one in a peridomestic site. Further research should be conducted to investigate the absolute contribution of humans and indoor transmission to the transmission cycle of CL. Inhabitants of endemic villages should be made aware that evening outdoor activities near caves may increase their exposure to infectious sand flies. Whether *P. orientalis* can breed and become infected at high altitudes should be further studied.

## 1. Background

A bite of a *Leishmania*-infected female phlebotomine sand fly vector can cause cutaneous leishmaniasis (CL). CL is a skin disease that manifests in lesions which predominantly occur on patients’ face and extremities and leave disfiguring scars after healing. It is the most common neglected tropical diseases (NTDs) involving the skin in Ethiopia, with a yearly estimate of 20,000 to 50,000 cases, although less than 800 were reported to WHO in 2021 [1,2]. The disease occurs on the slopes of the Ethiopian Great Rift Valley spanning from northwest to southwest and from south to east of the country, at altitudes ranging between 1600 and 2700 m [3]. Although there are a few well-described hotspots of CL [4,5,6,7,8], endemic areas are highly underreported and the distribution is more widespread than previously described.

Cutaneous leishmaniasis in Ethiopia, and some parts of Kenya, is mainly caused by *Leishmania aethiopica,* although a recent study in northern Ethiopia showed a few CL causes caused by *L. donovani* [9]. CL caused by *L. aethiopica* is generally severe and manifests in three forms: localized CL (LCL), mucocutaneous leishmaniasis (MCL) and diffuse cutaneous leishmaniasis (DCL).

The principal animal reservoirs for *L. aethiopica* are hyraxes (*Heterohyrax brucei* and *Procavia capensis*), which are abundant in rocky areas and caves [10]. Infection prevalence in these animals was found to be up to 20% [4,6]. Humans are also proven to efficiently transmit *L. aethiopica* to the vector [11]. *Phlebotomus (Larroussius) longipes* and *P. (La.) pedifer* have been identified as the main vectors of *L. aethiopica* in the north and south of the country, respectively [6]. Their breeding sites are rocky areas, caves, stony fences surrounding houses, gorges and tree cavities, where sand fly larvae feed on fecal materials of hyraxes [6,12,13]. In southern and central Ethiopia, research on the behavior of the vector has shown that their blood meal sources were mainly derived from humans, hyraxes and livestock, although the latter have never been found infected with *Leishmania* parasites. The vectors were demonstrated to be predominantly endophagic and mainly active around midnight [7,14].

The ecology and behavior of a vector are highly dependent on climate, topography and other environmental factors, which are different in northwestern Ethiopia compared to the center and south of the country. Yet, most information about the vector’s abundance, ecology and behavior originates from the latter two, questioning whether this can be generalized to the northwest, where transmission studies in CL endemic areas are scarce.

Patient records from the Leishmaniasis Research and Treatment Center (LRTC) in Gondar, northwest Ethiopia, demonstrate that there are many CL cases coming from both Gondar city and rural areas in the surroundings. There is no knowledge yet on how the transmission dynamics in the north and south or in urban and rural areas differ from each other, which should be properly investigated in order to design integrated vector control measures.

In this study, we assessed the spatial and temporal distribution of (infected) sand flies in an urban and rural endemic area of CL in northwestern Ethiopia. The findings will contribute to a better understanding of the disease’s transmission dynamics and design of control measures.

## 2. Methods

### 2.1. Description of Study Sites

This study was conducted in two sites that are known to be endemic of CL, based on the records of the LRTC in Gondar: an urban village, Addis-Alem, in Gondar city and a rural village, Gindmeteaye, in Lay-Armachiho woreda (district). Both are situated in the Amhara region in northwest Ethiopia, where Gondar is one of the major cities (Figure 1).

Addis-Alem is located at an elevation of approximately 2100 m at the border of Gondar city. It is an urban area, with a rocky cliff with hyraxes situated at one end of the village, nearby a mosque where people go to pray, also at night. Gindmeteaye lies at an elevation of about 2400 m at 29 km from Gondar. It is a rural area, mostly covered by farmland, where hyraxes and rocky areas are abundant throughout the village.

In both villages, household compounds are surrounded by stony fences with crevices in between. The study sites have a similar climate, with an annual rainfall of about 200–600 mm per year. The rainy season starts in May until the end of September and has cool temperatures ranging between 18 and 22 °C. In the dry season, from October to April, the temperature is generally higher, ranging from 22 to 28 °C.

### 2.2. Sand Fly Collections

Sand flies were collected during the dry season in February, March and April 2021 and during the wet season in May, July and August 2021 in both rural and urban villages. Each month, trapping was performed for five consecutive nights in the two study sites. In every village, six trapping sites were employed in three habitats; two indoors in houses of patients with active CL at the start of the study (identified by a physician), two peridomestic places outside a CL patient’s house and two in caves or rocky areas where hyraxes resided. In each collection site, trapping was carried out using one standard CDC miniature light trap (John W. Hock Company, Grainesville, FL, USA; Appendix A) set-up at 0.5 to 1 m above the ground, and five A4-format laminated sticky traps covered with sesame oil on both sides, placed against walls or within rock crevices. Traps were set-up at dusk and collected at dawn (Figure 2). The employed collection methods did not affect the environment. Collected sandflies were sorted and stored in absolute ethanol at −20 °C until further processing.

### 2.3. Sandfly Dissection and Morphological Identification

After collection, sand flies were sorted under the stereomicroscope according to sex, genus (*Phlebotomus* or *Sergentomyia*) and engorgement status (unfed/sugar fed, blood fed or gravid). Subsequently, the head and last segments of the abdomen were dissected and used for mounting in CMCP-10 high viscosity mountant (Polysciences, Hirschberg, Germany). All mounted sand flies were identified to species level using relevant morphological keys [17,18,19,20,21]. The thorax and remaining part of the abdomen were individually stored again in absolute ethanol at −20 °C for molecular analysis.

### 2.4. Leishmania Detection in Sandflies

All female *Phlebotomus* sand flies were molecularly screened for *Leishmania* parasites. Nucleic acid isolation of the individual sand flies’ thorax and abdomen was performed using a crude, in-house prepared extraction buffer, as described previously [13]. In each extraction batch of 22 samples, two negative extraction controls (lysis buffer only) and a positive extraction control (*Lutzomyia longipalpis*, laboratory-infected with *Leishmania major*) were added to check for contamination and extraction efficiency. Extracts were diluted with nuclease-free water, after which they were subjected to a PCR, targeting the 39 bp spliced-leader RNA (SL-RNA) gene, as described previously [22]. In each run, two positive PCR controls, two non-template controls and the extraction controls were added to assess the validity of the extraction and PCR run. Amplification was performed using the QuantStudio5 Real-Time PCR System (Applied Biosystems, ThermoFisher Scientific, Merelbeke, Belgium). Samples were considered positive if they had a cycle threshold value < 35 and a melting temperature (Tm) less than 0.5 °C different from the average Tm of the positive controls. The *P. longipes* specimens positive for SL-RNA were subjected to a PCR targeting the ITS-1 gene followed by a high resolution melt (HRM) curve analysis to determine the species of *Leishmania*. The assay was performed as described previously [23] and run on a Rotor Gene Q instrument (Qiagen, Antwerp, Belgium), including the DNA of cultivated *Leishmania* species as controls (*L. aethiopica* MHOM/ET/72/L127, *L. tropica* MHOM/PS/2003/LRC-L1022, and *L. donovani* MHOM/SD/2007/Ged4).

### 2.5. Data Analysis

Statistical analysis was performed in R Studio version 4.1.3 [24]. Chi-squares tests were employed to compare the sand fly abundance and blood feeding status between habitats, seasons and study sites. Binomial tests were used compare the difference in infection rate of *P. longipes* between habitats, seasons and study sites. The level of significance was *p* < 0.05.

## 3. Results

### 3.1. Species Typing

A total of 1161 sand flies were caught in the two study sites combined (Table 1), of which 838 (72%) were trapped with CDC light traps and the remaining 323 with sticky traps. Overall, 256 specimens (22%) belonged to the *Sergentomyia* genus and 905 (78%) were *Phlebotomus*. Among the *Sergentomyia*, we identified 156 (13%) *S. schwetzi*, 77 (7%) *S. clydei* and 23 (2%) *S. adleri.* Two species of *Phlebotomus* were identified, including 899 (77%) *P. longipes* and 6 *P. orientalis*. About half (430/899, 48%) of the *P. longipes* and two out of six *P. orientalis* were female. While the proportion of *P. longipes* captures in the rural and urban villages were similar, all *P. orientalis* sand flies were captured in the rural Gindmeteaye village.

### 3.2. Spatio-Temporal Sandfly Abundance

Of the 430 female *P. longipes* sand flies, 254 (59%) were found in the dry and 176 (41%) in the wet season (Table 2). Preference for habitats was significantly linked to seasonality (*p* < 0.001). While in the dry months, more (*p* < 0.001) sand flies were captured in outdoor habitats like rocky areas (37%) and peridomestic areas (34%), in the wet season, the majority (*p* = 0.001) of the female *P. longipes* sand flies were captured indoors (62%), with only 28% found in peridomestic are and 10% in rocky areas and caves. This seasonality in habitat preference was similar in both sites, with 63% of the female *P. longipes* and flies residing inside houses in the wet season in rural Gindmeteaye and 60% in urban Addis-Alem. Four male and one female *P. orientalis* were all captured inside case houses, and one specimen was collected in a peridomestic habitat. All six were found at the start of the rainy season in May.

### 3.3. Spatio-Temporal Variation in Sandfly Physiological Status

Among the 430 female *P. longipes*, 88 (20%) were blood-fed, 59 (14%) gravid and the remaining 283 (66%) were either sugar- or non-fed (Table 3). The proportion of blood-fed and gravid sand flies was higher in rural Gindmeteaye (28% and 18%, respectively) than in urban Addis-Alem, where 13% of the *P. longipes* specimens were blood-fed and 10% were gravid (Table 4). The number of blood-fed specimens was significantly higher in the wet (55/176, 31%) compared to the dry season (33/254, 13%, *p* < 0.001, Table 3). Likewise, significantly more gravid sand flies were found in the wet (38/176, 22%) than the dry season (21/254, 8%, *p* < 0.001). This trend was similar for all three habitats. The proportions of blood-fed and gravid sand flies were highest indoors, with 24% (44/182) and 16% (29/182), respectively, followed by 19% and 13% in peridomestic areas and 16% and 11% in caves/rocky areas, although these differences were not significant (*p* = 0.235 for blood-fed and *p* = 0.463 for gravid). Of the two *P. orientalis* female sand flies, the specimen found inside the CL case house was blood-fed; the other one was non-fed.

### 3.4. Leishmania Infection

Of the 430 individual *P. longipes* sand flies tested by PCR for *Leishmania*, 18 (4%) were positive (Table 4). The species of *Leishmania* could not be successfully determined. Infection prevalence was significantly lower (*p* = 0.005) in rural Gindmeteaye with 2% (4/202), compared to 6% (14/228) in urban Addis-Alem. The proportion of infected *P. longipes* was highest in caves, where eight out of 110 (7%, *p* = 0.022) were positive for *Leishmania*, followed by 4% (5/138) for peridomestic areas and 3% (5/182) indoors. The proportion of infected sand flies was higher in the dry (6%, *p* < 0.001) than the wet (2%) season. Ten of the positive sand flies were non-fed, five were blood-fed and three were gravid. The two female *P. orientalis* sand flies were both negative for *Leishmania* PCR.

## 4. Discussion

Our study is the first eco-epidemiological study investigating the transmission of CL in northwest Ethiopia. We demonstrate that transmission is ongoing in urban Addis-Alem, in the suburbs of Gondar city, which lies at the very suitable altitude, for *P. longipes*, of 2100 m and has caves and stony fences around houses. While most known hotspots of CL are described to be in rural areas, another study showed that CL in Ethiopia occurs in the areas surrounding Addis Ababa, where a cliff and hyraxes are present, indicating the importance of the animal reservoir [4]. Based on records from the LRTC in Gondar, cases are indeed also coming from Gondar city. Future studies should investigate whether infected sand flies also occur more towards the center of the city, where hyraxes are not present to determine what the role of human hosts in the transmission cycle is when the animal reservoir is not present.

Majority of the sand flies in our study were *P. longipes*, which corresponds with previous entomological research conducted in Gondar and Lay-Armachiho (the district in which Gindmeteaye village is situated). However, these authors also collected three *P. gibiensis* specimens in both sites [25], which were not found in our study.

Similar to findings from southern Ethiopia, most female *P. longipes* in urban Addis-Alem were captured in the dry season [13]. However, in rural Gindmeteaye, *P. longipes* was most abundant in the wet season. This could be because sand flies were predominantly found indoors here, whereas, in urban Addis-Alem, the abundance was more equal throughout the different habitats. Although no objective records are available from our study, field observations indicate that housing structures in urban Addis-Alem were better than in rural Gindmeteaye, where there were more gaps and cracks in the walls and roof. The better housing structures could potentially have prevented sand flies from entering the houses for feeding and resting [26]. For malaria, improving housing structures and screening doors and windows was proven effective at protecting humans from indoor mosquito vector bites and reducing their risk of infection [27]. More data are needed to objectively investigate whether improved housing structures are related to lower numbers of indoor sand fly vectors, to determine whether screening could also be an effective control measure for endophilic sand fly vectors.

More female *P. longipes* were captured indoors in the wet compared to the dry season in both sites, indicating that the vector seeks shelter indoors when it is rainy. This important finding could not be demonstrated in the south, where sand fly collection was not performed indoors, leading to a biased low abundance of sand flies in the wet season [13]. Possibly, if indoor collections would have been performed in that study, *P. pedifer* would have been more abundant indoor in the wet season as well. This indicates that, potentially, vector control strategies should be different in the wet and dry season. Importantly, the proportion of blood-fed sand flies was found to be the highest indoors, indicating the potential for indoor transmission, as was previously described in the south, where most blood-fed sand flies were found indoors and fed on humans. However, blood meal sources were not determined in our study and could be different from the south. Likewise, the proportion of gravid sand flies was highest indoors, which could indicate that the vector is also breeding indoors or in peridomestic areas, in which case the dung of livestock could serve as a breeding site. The presence of animal dung inside household compounds was recently identified as a risk factor for CL (manuscript in preparation). Hence, further research is required to determine whether sand flies can breed in animal dung, which can be determined based on the rotation of genitalia of male sand flies [28].

The overall *Leishmania* infection prevalence of *P. longipes* in our study was 4%, which is in line with findings from other endemic sites in the country. Infection rates were highest in urban Addis-Alem, despite the fact that, based on hospital records, more patients come from Gindmeteaye. Additionally despite the fact that most blood-fed and gravid sand flies were found inside houses, most infected *P. longipes* were found in caves. This is in line with previous findings from southern Ethiopia, where significantly more infected sand flies were found in caves [7,13].

In contrast to the proportion of blood-fed and gravid sand flies, which was found to be highest indoors in the wet season, the proportion of infected sand flies was highest in the dry season, when sand flies are mostly found in caves and peridomestic habitats, confirming that hyraxes play an important role in transmission [4,6,13,29]. This also corresponds with risk factor studies indicating that undertaking outdoor activities (e.g., fetching water or wood and herding animals) near hyrax habitats or caves in particular is a significant risk factor (manuscript in preparation). Hence, communities should be made aware of the risks of such activities after dusk.

Overall, more evidence is required on the absolute contributions of hyraxes and humans to the transmission of CL in both seasons. Furthermore, it should be investigated whether humans have the highest risk of infection inside their dwellings or during outdoor activities near caves. This would enable the design of effective integrated control measures for the control of CL in northwest Ethiopia.

Besides *P. longipes*, we also found *P. orientalis* in rural Gindmeteaye. *Phlebotomus orientalis* is the vector for *L. donovani*, causing visceral leishmaniasis (VL) in northwest Ethiopia and Sudan. It commonly occurs in the lowlands where temperatures are high, and its presence is associated with habitats like vertisols, *Acacia* and *Balanites* trees [30]. Our study reports the highest elevation (2400 m) from which *P. orientalis* has ever been collected, with previous records at high altitudes of 1800 m in Addis Zemen [31] and 2000 m in Libo-Kemekem [32,33] in Ethiopia. The *P. orientalis* specimens in our study were captured in May, which is the hottest month of the year in the study area, yet the typical habitats are not present in Gindmeteaye. Five out of six specimens were found indoors. Although only few specimens were captured, a potential explanation of this species occurring at such high altitude could be climate change, allowing the species to survive here in the hottest months but indoors only. Such changes can result in the altered behavior of the vector as it adapts to new climatic conditions. Further studies need to explore whether the species can breed and become infected at high elevations, as that could have a huge impact on immunologically naive populations, potentially leading to outbreaks of VL. There are some confirmed VL cases presenting at the LRTC in Gondar who do not have any travel history to lowland areas where the disease is endemic. This is currently being further investigated. Moreover, recent evidence indicates that CL caused by *L. donovani* occurs in Ethiopia as well [9], although no whole genome sequencing was performed to assess whether this could have been a hybrid between *L. donovani* and *L. aethiopica.* This hybridization could have occurred in the midgut of *P. orientalis* [34,35]. One should further investigate whether *P. orientalis* is a permissive vector and could potentially lead to the formation of hybrid species, which could be a threat to public health in Ethiopia and beyond.

In this study, we trapped sand flies in both the wet and dry season, in indoor and outdoor habitats. The limitations are that we did not isolate the *Leishmania* parasites or determine the *Leishmania* species and blood meal sources of the sand flies. Such information would be valuable to further investigate which hosts play an important role in transmission. Moreover, information on the weather conditions (temperature, rain fall) and housing structures is lacking, so the hypothesis that good housing structures could potentially prevent sand flies from entering houses cannot objectively be substantiated. Generally, a follow-up study in the region should employ a One Health approach, so that a holistic overview of the situation can provide the necessary insights to develop a successful control program for CL.

## 5. Conclusions

This eco-epidemiological investigation in CL-endemic rural Gindmeteaye and urban Addis-Alem in northwest Ethiopia revealed that *P. longipes* was mostly found outdoors, in caves and peridomestic ecotypes, in the dry season. Likewise, *Leishmania* prevalence in *P. longipes* was highest in the dry season, in caves. On the contrary, in the wet season, most blood-fed and gravid sand flies were captured indoors, with a lower proportion infected. Further research is required to investigate the absolute contribution of humans and hyraxes to the CL transmission cycle and determine whether the highest risk of infection is indoors or outdoors. Furthermore, we showed that *P. orientalis* occurred in CL-endemic areas, although in low numbers. It should be investigated whether they can breed and become infectious as such high altitudes, and whether they are permissive vectors, which could potentially lead to the formation of hybrid species.

## Figures and Tables

**Figure 1 tropicalmed-09-00052-f001:**
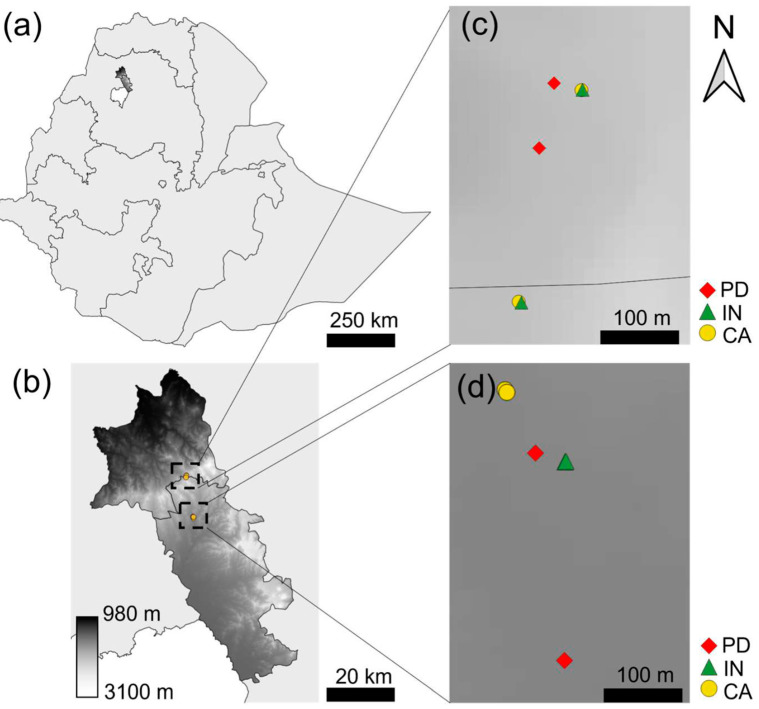
Map of study sites [15,16]: (**a**) Ethiopia, with, in the northwestern Amhara region the two study districts. (**b**) Upper district is Lay-Armachiho, where our rural study village Gindmeteaye is situated at around 2400 m, and the lower district is Gondar Zuria, where our urban study village Addis-Alem is located at about 2100 m. (**c**) Zoom-in of sampling points in rural Gindmeteaye. (**d**) Zoom-in of sampling points in urban Addis-Alem. PD: peridomestic, IN: indoor and CA: cave collections.

**Figure 2 tropicalmed-09-00052-f002:**
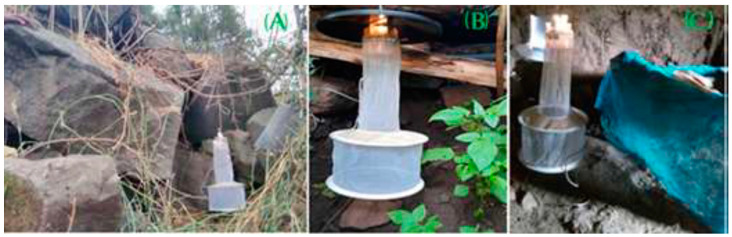
Sandfly collection from three different habitats: (**A**) caves, (**B**) peridomestic (outdoor) and (**C**) indoor. *Pictures: Wondmeneh Jemberie*.

**Table 1 tropicalmed-09-00052-t001:** Species distribution of sand flies in urban Addis-Alem and rural Gindmeteaye. Proportions were calculated with the total number of sand flies of a certain sex at a certain site as the denominator (last row).

		Gindmeteaye			Addis-Alem			Overall		
		Rural			Urban					
		Female	Male	Total	Female	Male	Total	Female	Male	Total
		N (%)	N (%)	N (%)	N (%)	N (%)	N (%)	N (%)	N (%)	N (%)
*Phlebotomus*										
	*P. longipes*	203	266	466	227	203	429	430	469	899
		(80.2)	(89.0)	(84.9)	(69.8)	(71.0)	(70.1)	(74.7)	(80.2)	(76.6)
	*P. orientalis*	2	4	6	0	0	0	2	4	6
		(0.8)	(1.3)	(1.1)	(0.0)	(0.0)	(0.0)	(0.3)	(0.7)	(0.5)
*Sergentomyia*										
	*S. adleri*	7	9	16	1	9	10	5	18	23
		(2.8)	(3.0)	(2.9)	(0.3)	(3.1)	(1.6)	(0.9)	(3.1)	(2.0)
	*S. clydie*	17	17	34	7	34	41	24	51	77
		(6.7)	(5.7)	(6.2)	(2.2)	(11.9)	(6.7)	(4.2)	(8.8)	(6.6)
	*S. schwezi*	24	3	27	90	40	132	113	43	156
		(9.5)	(1.0)	(4.9)	(27.7)	(14.0)	(21.6)	(19.6)	(7.4)	(13.4)
Overall total		253	299	549	325	286	612	576	585	1161

**Table 2 tropicalmed-09-00052-t002:** Seasonal distribution of female *Phlebotomus longipes* sand flies in urban Addis-Alem and rural Gindmeteaye in indoor and outdoor habitats. Percentages not in bold are the proportion of the sand flies within a particular habitat in a certain season, with the total number of sand flies in that season as the denominator. The percentages in bold are the proportions of sand flies captured in that certain season, with the total number of sand flies captured in that season as the denominator.

		Dry	Wet	Both Seasons
Addis-Alem (urban)				
	Indoor	49 (28%)	30 (60%)	79 (35%)
	Peridomestic	61 (34%)	13 (26%)	74 (32%)
	Rocky/cave	68 (38%)	7 (14%)	75 (33%)
	Total	178 (78%)	50 (22%)	228
Gindmeteaye (rural)				
	Indoor	24 (31%)	79 (63%)	103 (51%)
	Peridomestic	27 (36%)	37 (29%)	64 (32%)
	Rocky/cave	25 (33%)	10 (8%)	35 (17%)
	Total	76 (38%)	126 (62%)	202
Villages combined				
	Indoor	73 (29%)	109 (62%)	182 (42%)
	Peridomestic	88 (34%)	50 (28%)	138 (32%)
	Rocky/cave	93 (37%)	17 (10%)	110 (26%)
	Total	254 (59%)	176 (41%)	430

**Table 3 tropicalmed-09-00052-t003:** Spatio-temporal blood feeding status of female *Phlebotomus longipes*.

		Dry	Wet	Both Seasons
Indoor				
	Blood-fed	15 (20%)	29 (27%)	44 (24%)
	Gravid	5 (7%)	24 (22%)	29 (16%)
	Non-fed/sugar-fed	53 (73%)	56 (51%)	109 (60%)
	Total	73	109	182
Peridomestic				
	Blood-fed	6 (7%)	20 (40%)	26 (19%)
	Gravid	8 (9%)	10 (20%)	18 (13%)
	Non-fed/sugar-fed	74 (84%)	20 (40%)	94 (68%)
	Total	88	50	138
Caves/rocky areas				
	Blood-fed	12 (13%)	6 (35%)	18 (16%)
	Gravid	8 (9%)	4 (24%)	12 (11%)
	Non-fed/sugar-fed	73 (78%)	7 (41%)	80 (73%)
	Total	93	17	110
All habitats combined				
	Blood-fed	33 (13%)	55 (31%)	88 (20%)
	Gravid	21 (8%)	38 (22%)	59 (14%)
	Non-fed/sugar-fed	200 (79%)	83 (47%)	283 (66%)
	Total	254	176	430

**Table 4 tropicalmed-09-00052-t004:** Spatio-temporal infection status of female *Phlebotomus longipes*.

		Total	Infected
Trap site			
	Addis Alem (urban)	228	14 (6%)
	Gindmeteaye (rural)	202	4 (2%)
Habitat			
	Caves	110	8 (7%)
	Peridomestic	138	5 (4%)
	Indoors	182	5 (3%)
Season			
	Dry	254	15 (6%)
	Wet	176	3 (2%)
Blood feeding status			
	Non-fed	283	10 (4%)
	Blood-fed	88	5 (6%)
	Gravid	59	3 (5%)

## Data Availability

Data will be made available upon request to the authors.

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
