# Peer review of "Ecology and Infection Status of Sand Flies in Rural and Urban Cutaneous Leishmaniasis Endemic Areas in Northwest Ethiopia"

_tropicalmed, 2024, doi:10.3390/tropicalmed9030052_

Round 1

Reviewer 1 Report

Comments and Suggestions for Authors

The manuscript presents the results of entomological and parasitological investigations in endemic areas of CL in northern Ethiopia. I recommend the publication after minor revision:

- Abstract must be structured as: Background/objective/MM/Results and Conclusion 

- Introduction section: Epidemiological status of CL in the study area must be well presented (leishmaniasis forms, cases, proven vectors, reservoir...)

The objective must be reviewed and adapted 

- Results section: Present and discuss the sand fly distribution according to traping techniques (CDC LT or sticky paper)

Please present the results of sand fly infection in the table  

Author Response

We thank the reviewer for the valuable comments and have tried to address them as good as possible and provided responses to the requests here below.

The manuscript presents the results of entomological and parasitological investigations in endemic areas of CL in northern Ethiopia. I recommend the publication after minor revision:

Abstract must be structured as: Background/objective/MM/Results and Conclusion

We think that is how we have structured it, although we noticed from the guidelines and other published articles in MDPI TMID that there is no subheadings in the abstract, so we have left it out.

Introduction section: Epidemiological status of CL in the study area must be well presented (leishmaniasis forms, cases, proven vectors, reservoir...)

To clarify the situation to the reader, we have added some information to the background to elaborate on the epidemiology of CL in Ethiopia, including the forms of leishmaniasis, burden and some additional information on the vector and reservoir.  

The objective must be reviewed and adapted 

We respectfully would like to request what the reviewer would particularly like to adapt. Can you please be a bit more specific so that we can consider the requested changes?

Results section: Present and discuss the sand fly distribution according to traping techniques (CDC LT or sticky paper)

In the first section of the results we have added the proportion of sand flies that were trapped by CDC LT and sticky traps.

Please present the results of sand fly infection in the table  

We have presented the infection results per trapping site, habitat, season and blood feeding status of captured P. longipes sand flies in a new Table 4 in the manuscript as requested.

Reviewer 2 Report

Comments and Suggestions for Authors

Title:

Ecology and Infection Status of Sand Flies in Rural and Urban Cutaneous Leishmaniasis Endemic Areas in Northwest Ethiopia.

Manuscript ID: 2767040

ABSTRACT

-          The second sentence could be changed to: ¨However, no studies have been conducted to investigate the transmission dynamics of CL, despite its high endemicity…¨

-           Rewrite the phrase ¨yet six were identified as P. orientalis¨. Percent could be included.

-          Almost a sentence that mentioned rest of identified phlebotominae species could be positive.

-          In conclusion, the impact of study could be include.

INTRODUCTION

-          In the first sentence should be clarify that infection of phlebotomine is with Leishmania parasite, which cause the infection, not a bite.

MATERIALS AND METHODS

-          Studied area was described. However, could be relevant some epidemiological data related with leishmaniasis diseases in humans.

-          In figure 2, specify if pictures were taken by authors.

-          Include the commercial house, city and country for each reactive and equipment.

-          Specify if in the study area insecticides have been used and that the performed collection did not affect the environment (if you have an ethical code or statement could be added).

RESULTS

-          Include statistical analysis in Table 2 and 3.

-          Leishmania positive samples were mentioned. However, I would like to know if some quantitative analysis of parasite burden could be included.

-          Only positivity of Leishmania was mentioned, however, identification of specie could be positive. Please, can include some comments into the manuscript in parallel to mention as a limitation.

GENERAL COMMENTS

-          Review scientific names and abbreviations in complete manuscript.

Comments on the Quality of English Language

-

Author Response

We thank the reviewer for the valuable comments and have tried to address them as good as possible and provided responses to the requests here below.

Title: Ecology and Infection Status of Sand Flies in Rural and Urban Cutaneous Leishmaniasis Endemic Areas in Northwest Ethiopia. Manuscript ID: 2767040 

ABSTRACT

The second sentence could be changed to: ¨However, no studies have been conducted to investigate the transmission dynamics of CL, despite its high endemicity…¨

We have removed the “here” in the sentence and adapted it as the reviewer suggested, but we left out the word “however” because we do not think that it would improve the sentence.

Rewrite the phrase ¨yet six were identified as P. orientalis¨. Percent could be included. Almost a sentence that mentioned rest of identified phlebotominae species could be positive.

We have revised the sentence and added the percentage and other species as follows: “Of 1161 sand flies, majority (77%) were P. longipes, six (0.5%) were P. orientalis and the remaining were Sergentomyia.” We think that mentioning specifically which Sergentomyia species we found is not sufficiently relevant to be added in the abstract.

In conclusion, the impact of study could be include. 

We thank the reviewer for this comment, although it is not entirely clear to us what is requested to be added here. We have already mentioned that there is a scientific impact, as more research needs to be done to make guidelines, and additionally that inhabitants should be made aware of the risk of exposure to infected sand flies nearby caves in the evening. If the reviewer can be a bit more specific on what exactly should be added here, we are happy to consider it.

INTRODUCTION

In the first sentence should be clarify that infection of phlebotomine is with Leishmania parasite, which cause the infection, not a bite.

We have revised the sentence as recommended by the reviewer: “A bite of a Leishmania-infected female phlebotomine sand fly vector can cause cutaneous leishmaniasis (CL)“

MATERIALS AND METHODS

Studied area was described. However, could be relevant some epidemiological data related with leishmaniasis diseases in humans.

We appreciate the reviewer’s comment. However, there is very little epidemiological data available about the site. We have added a sentence to the revised version of the manuscript that the sites were selected based on records from the LRTC in Gondar, where it was observed that patients frequently come from these areas.

In figure 2, specify if pictures were taken by authors.

      As recommended, we added to the figure legend that the pictures of Figure 2 were collected by the lead author, Wondmeneh Jemberie.

Include the commercial house, city and country for each reactive and equipment.

We have added the brand name, city and country to all reagents and equipment listed in the manuscript as requested by the reviewer.

Specify if in the study area insecticides have been used and that the performed collection did not affect the environment (if you have an ethical code or statement could be added).

We respect the reviewer’s comment. We have added in the methods section 2.2 Sand fly collection that the employed trapping methods did not affect the environment. We feel however, that adding something about the use of insecticides in the study area does not add much to the manuscript and hope that the reviewer accepts that we decided to leave this out.  

RESULTS

Include statistical analysis in Table 2 and 3.

We thank the reviewer for this valuable comment. We agree with the question, however, as we have used statistics in multiple directs in the tables, we feel it would make the tables more complicated and hence prefer to present the statistics only in the manuscript text.

Leishmania positive samples were mentioned. However, I would like to know if some quantitative analysis of parasite burden could be included.

We thank the reviewer for this comment. If we understand the suggestion correctly, the reviewer is asking to add a table in which the exact values of infected sand flies are presented. Therefore we have added Table 4 to the manuscript in which we present the infection prevalence per trapping site, habitat, season and blood feeding status. Kindly let us know if we have misinterpreted the reviewer’s request.

Only positivity of Leishmania was mentioned, however, identification of species could be positive. Please, can include some comments into the manuscript in parallel to mention as a limitation.

We agree with the reviewer’s request to know the species of Leishmania that caused the infection in sand flies, especially because indeed recently a report has come out that CL caused by L. donovani was detected in northern Ethiopia and that we found the L. donovani vector in the highlands. Recently, a PCR was implemented at the LRTC in Gondar which can determine the species of Leishmania easily. Thus, we did extra testing with our study samples and subjected the ones positive for SL-RNA to a qPCR targeting ITS-1 followed by a high-resolution melting curve analysis. Unfortunately we have not been able to determine the Leishmania species with this method, probably because of the fact that due to the conflict there have been a lot of power interruptions at the university of Gondar, which affected the specimens stored in the freezers. The methods that were used to determine the species and results were added to the manuscript. We have added the lack of species typing in our work as a limitation of the study in the discussion section.    

GENERAL COMMENTS

Review scientific names and abbreviations in complete manuscript.

We have reviewed the text and checked for all scientific names and abbreviations and tried to put all in italics and full, respectively. If we have overseen some, please notify us and we will adapt it accordingly.

Reviewer 3 Report

Comments and Suggestions for Authors

The study titled “Ecology and Infection Status of Sand Flies in Rural and Urban Cutaneous Leishmaniasis Endemic Areas in Northwest Ethiopia.” Is a short study with exciting results. The authors identify the presence of the parasites wherein Leishmania parasites are usually not recorded at high altitudes. The findings are interesting and good contribution to the field though I would recommend the authors add more results and make it an extensive study i.e. identify species ..etc. I have also suggested some comments for improving the manuscript.

1.      Figure 1: It will be good to show the altitude contours in the map for the readers to understand the elevation. Also, provide the location of the sample collection as red spots on the graph.

2.      Section 2.2: mention the altitudes of the location to compare the results.

3.      Table 2: Check the percentages in the table. For example, Addis-Alem: Indoor: Dry is 49/175 which is 27% rather than 25; for wet it is 60% rather than 53. Check for calculation errors throughout the manuscript and recalculate the significance.

4.      The proportion of infected sandflies that were blood-fed in different regions should also mentioned in section 3.4.

5.      The results of parasite infection detection in different regions and seasons should be tabulated too.

6.      The dependence of the sandflies on the temperature of the regions should be discussed in detail.

7.      Line 274: did the authors mean “native” population?

8.      The authors should also add a paragraph on measures to improve the situation and policy changes with respect to One Health.

Comments on the Quality of English Language

The manuscript required minor language editing. 

Author Response

The study titled “Ecology and Infection Status of Sand Flies in Rural and Urban Cutaneous Leishmaniasis Endemic Areas in Northwest Ethiopia.” Is a short study with exciting results. The authors identify the presence of the parasites wherein Leishmania parasites are usually not recorded at high altitudes. The findings are interesting and good contribution to the field though I would recommend the authors add more results and make it an extensive study i.e. identify species etc. I have also suggested some comments for improving the manuscript.

We thank the reviewer for the valuable, constructive comments and have tried to address them as good as possible to improve our manuscript and provided responses to the requests here below.

Figure 1: It will be good to show the altitude contours in the map for the readers to understand the elevation. Also, provide the location of the sample collection as red spots on the graph.

We thank the reviewer for this valuable comment which indeed improves the figure a lot and makes it better to understand the trapping sites for the reader. We have adapted Figure 1 by adding the altitude gradient and presenting the trapping sites with different icons and colors.

Section 2.2: mention the altitudes of the location to compare the results.

The altitudes of the trapping areas are already mentioned under section 2.1 Description of the study sites, and we are not sure where and why to repeat this in section 2.2. If the reviewer can clarify this, we can consider to include it. We have, however, added the altitudes still in the legend of Figure 1 so that this immediately becomes clear when looking at the figure only that the altitudes were different.

Table 2: Check the percentages in the table. For example, Addis-Alem: Indoor: Dry is 49/175 which is 27% rather than 25; for wet it is 60% rather than 53. Check for calculation errors throughout the manuscript and recalculate the significance.

We are very grateful for the reviewer to spot this mistake. We have corrected the proportions in the table and adapted the text accordingly in the revised version of the manuscript. We also reviewed the rest of the manuscript for errors and checked the level of significance.

The proportion of infected sandflies that were blood-fed in different regions should also mentioned in section 3.4.

We agree with the reviewer’s comment that this is still valuable information to be added to the manuscript. By this subset analysis we have seen that the proportion blood fed and gravid sand flies was higher in the rural than the urban site. We have added this to the manuscript in the text under section 3.3 and have added it also in Supplementary Table S1 so that readers can find the data in a table too.

The results of parasite infection detection in different regions and seasons should be tabulated too.

We agree with the reviewer’s valuable comment and have added the infection prevalence per trapping site, habitat, season and blood feeding status in Table 4 in the revised version of the manuscript.

The dependence of the sandflies on the temperature of the regions should be discussed in detail.

We are in agreement with the suggestion raised by the reviewer that the sand fly abundance and maybe even its feeding behavior and infection rate is probably related to the temperature and other weather conditions. However, we do not have these data available, so feel that discussion this in detail in the manuscript might be an overinterpretation of the results. We have added this as a limited of the study in the discussion of the revised version of the manuscript.

Line 274: did the authors mean “native” population?

      We understand that this wording needs to be clarified. We actually meant that it would be a risk to populations that have never been exposed to the parasite before, because it could lead to outbreaks. Therefore we have revised the wording now to “immunologically naïve population”.

The authors should also add a paragraph on measures to improve the situation and policy changes with respect to One Health.

We thank the reviewer for this suggestion. We have added a section at the final part of the discussion as follows: “Generally, a follow-up study in the region should employ a One health approach, so that a holistic overview of the transmission dynamics can provide the necessary insights to develop a successful control program for CL.” We hope that this is about what the reviewer is expecting. If not, kindly let us know.  

Reviewer 4 Report

Comments and Suggestions for Authors

Congratulations for the well structured study: "Ecology and Infection Status of Sand Flies in Rural and Urban Cutaneous Leishmaniasis Endemic Areas in Northwest Ethiopia", it appears of interest in the specific field of application reporting data concerning sand flies distribution, species and infectious Leishmania prevalences in an endemic focus for cutaneous leishmaniasis caused by Leishmania aethiopica, trying to clarify for the first time transmission dynamics in this regions. By the way further studies are required to achieve the goal by including wild and domestic reservoirs screening in the study area. Title reviewing is strongly suggested.

Comments on the Quality of English Language

Only minor revisin concerning English Language

Author Response

We thank the reviewer for the valuable comments and have tried to address them as good as possible and provided responses to the requests here below.

Congratulations for the well structured study: "Ecology and Infection Status of Sand Flies in Rural and Urban Cutaneous Leishmaniasis Endemic Areas in Northwest Ethiopia", it appears of interest in the specific field of application reporting data concerning sand flies distribution, species and infectious Leishmania prevalences in an endemic focus for cutaneous leishmaniasis caused by Leishmania aethiopica, trying to clarify for the first time transmission dynamics in this regions. By the way further studies are required to achieve the goal by including wild and domestic reservoirs screening in the study area. Title reviewing is strongly suggested.

We thank the reviewer for the positive feedback. We agree that studies on wild and domestic animal reservoirs should be done in the future. We have added this valuable comment in the discussion section of our revised manuscript. With regard to the comment on the title, we would like to know from the reviewer what exactly would need to be changed in order to improve it. If the reviewer can further explain this, we can consider to alter it. Thank you.

Round 2

Reviewer 3 Report

Comments and Suggestions for Authors

The authors have addressed all the comments satisfactorily. 

Author Response

We thank the reviewer for the valuable comments